

# Antimicrobial activity of Tachyplesin 1 against *Burkholderia pseudomallei*: an in vitro and in silico approach

Lyn-Fay Lee[1], Vanitha Mariappan[1], Kumutha Malar Vellasamy[1], Vannajan Sanghiran Lee[2] and Jamuna Vadivelu[1]

[1] Department of Medical Microbiology, Faculty of Medicine, University of Malaya, Kuala Lumpur, Malaysia
[2] Department of Chemistry, Faculty of Science, University of Malaya, Kuala Lumpur, Malaysia

## ABSTRACT

*Burkholderia pseudomallei*, the causative agent of melioidosis, is intrinsically resistant to many conventional antibiotics. Therefore, alternative antimicrobial agents such as antimicrobial peptides (AMPs) are extensively studied to combat this issue. Our study aims to identify and understand the mode of action of the potential AMP(s) that are effective against *B. pseudomallei* in both planktonic and biofilm state as well as to predict the possible binding targets on using in vitro and in silico approaches. In the in vitro study, 11 AMPs were tested against 100 *B. pseudomallei* isolates for planktonic cell susceptibility, where LL-37, and PG1, demonstrated 100.0% susceptibility and TP1 demonstrated 83% susceptibility. Since the *B. pseudomallei* activity was reported on LL-37 and PG1, TP1 was selected for further investigation. TP1 inhibited *B. pseudomallei* cells at 61.69 $\mu$M, and membrane blebbing was observed using scanning electron microscopy. Moreover, TP1 inhibited *B. pseudomallei* cell growth, reaching bactericidal endpoint within 2 h post exposure as compared to ceftazidime (CAZ) (8 h). Furthermore, TP1 was shown to suppress the growth of *B. pseudomallei* cells in biofilm state at concentrations above 221 $\mu$M. However, TP1 was cytotoxic to the mammalian cell lines tested. In the in silico study, molecular docking revealed that TP1 demonstrated a strong interaction to the common peptide or inhibitor binding targets for lipopolysaccharide of *Escherichia coli*, as well as autolysin, pneumolysin, and pneumococcal surface protein A (PspA) of *Streptococcus pneumoniae*. Homology modelled *B. pseudomallei* PspA protein (YDP) also showed a favourable binding with a strong electrostatic contribution and nine hydrogen bonds. In conclusion, TP1 demonstrated a good potential as an anti-*B. pseudomallei* agent.

# INTRODUCTION

*Burkholderia pseudomallei*, the causative agent for melioidosis, is commonly found in the soil and water of Southeast Asia, and Northern Australia (*Cheng & Currie, 2005*). Due to the ease of dissemination where infection can be acquired either through an open wound, inhalation, or ingestion (*Puthucheary, 2009*), coupled with the genomic

Corresponding author
Jamuna Vadivelu,
jamuna@um.edu.my

plasticity of the bacteria (*Holden et al., 2004*; *Schweizer, 2012*), it is labelled as a potential biological warfare agent and classified as a category B bio threat agent by the Centers for Disease Control and Prevention where specific enhancements are required for diagnostic capacity and disease surveillance (*Stöckel et al., 2015*). Currently, melioidosis is treated using parenteral therapy of ceftazidime (CAZ) followed by prophylactic therapy of co-trimoxizole (*Dance, 2014*). Over the years, *B. pseudomallei* has been reported to resist the commonly used antibiotics (increased usage of CAZ and amoxicillin/clavulanic acid in treatment), and also to the ability to form biofilm in vitro and in vivo (*Schweizer, 2012*). To exacerbate the problem, incomplete treatment resulted in a high rate of melioidosis relapse. Worst of all, despite appropriate antimicrobial therapy, mortality from melioidosis septic shock remained high (*Wiersinga et al., 2006*). As such, there is a need to consider alternative antimicrobial agents, one of which is the antimicrobial peptides (AMPs) (*Rotem & Mor, 2009*).

AMPs, both naturally occurring and synthetic, have considerable advantages for therapeutic applications, including broad-spectrum activity, rapid onset of activity and relatively low possibility of emergence of resistance (*Seo et al., 2012*). Furthermore, they act on slow-growing or even non-growing bacteria due to the ability to permeablise and form pores within the cytoplasmic membrane (*Batoni et al., 2011*). Groups of AMP's i.e., defensins, cathelicidins, and dermicins have previously been reported to show potential against various pathogens (*Wiesner & Vilcinskas, 2010*). However, to date, only a few cathelicidin AMPs have been reported to be effective against *B. pseudomallei*, including LL-37 (*Kanthawong et al., 2012*), protegrin 1 (PG1) (*Sim et al., 2011*), bovine lactoferrin (*Puknun et al., 2013*), phospholipase A2 inhibitors (*Samy et al., 2015*), and sheep cathelicidin (SMAP-29) (*Blower, Barksdale & van Hoek, 2015*).

To compliment in vitro investigations, in silico investigations were incorporated to elucidate the mechanism of action of the potential antimicrobial compounds. As structures of more protein targets become available through crystallography, nuclear magnetic resonance (NMR) and bioinformatics methods, there is an increasing demand for computational tools that can identify and analyse active sites and suggest potential drug molecules that can specifically bind to these sites. This gives rise to various docking programs such as DOCK, GLIDE, and Surfle which predict the preferred orientation of a drug (ligand) molecule to a receptor molecule when bound to each other to form a stable complex (*Kitchen et al., 2004*; *Lengauer & Rarey, 1996*) and also the strength of association or binding affinity between both the molecules (*Cross et al., 2009*). The in silico methods allow drug-ligand interaction studies to be performed in shorter time periods and aids the designing of better therapeutic compounds (*Gupta et al., 2013*; *Kapetanovic, 2008*). Incorporation of both in vivo and in silico techniques has been widely used in the recent years. *Le et al. (2015)* performed toxicity studies together with molecular docking to predict possible binding targets of AMP DM$_3$ using Autodock Vina where a strong affinity was demonstrated towards three targets; *Streptococcus pneumoniae* virulence factors; autolysin, pneumolysin, and pneumococcal surface protein A (PspA). In another study by *Sarojini et al. (2010)*, in silico molecular docking using Autodock 3.0 was carried out to support the in vitro antibacterial and antifungal

activity as well as to determine the orientation of the 2,5-dichloro thienyl substituted thiazole derivatives (4a–4d) bound in the active site of 3-deoxy-D-glucosamine (GlcN)-6-P synthase (target for antifungals). Similar use of in silico molecular docking to compliment the findings of the in vitro/in vivo experiments and elucidate the antimicrobial-protein interaction was also been reported in a study by in *Alves et al. (2013)* on the antimicrobial properties of mushroom phenolic compounds and *Al-Sohaibani & Murugan (2012)* on the inhibition effects of *Salvadora persica* on *Streptococcus mutans*.

Taking into account the effectiveness of AMPs to inhibit potentially antibiotic resistant bacteria, our study aimed to identify the potential AMP(s) that are effective against *B. pseudomallei* in both planktonic and biofilm state and to predict the possible binding targets on *B. pseudomallei* protein structures. Overall, there were two approaches in this study; in vitro and in silico. Initially, the in vitro experiments were executed to identify the potential AMP(s) that are effective against *B. pseudomallei* in both planktonic and biofilm state. Based on those observations, the in silico experiments were then carried out on the selected AMP to predict the possible binding targets on the available *B. pseudomallei* protein structures. Both of these approaches will assist in elucidating the action of the AMP on *B. pseudomallei* and contribute to the development of AMPs as an alternative anti-*B. pseudomallei* agent.

## MATERIALS AND METHODS

### Bacteria strains and growth conditions

A total of 90 *B. pseudomallei* clinical isolates from melioidosis patients (archive collection of University of Malaya Medical Centre (UMMC), Kuala Lumpur and Hospital Tengku Ampuan Afzan, Pahang) and 10 non-clinical isolates (one environmental and nine animal) were included in this study. All isolates were identified as *B. pseudomallei* using routine phenotypic characterization including selective growth on Ashdown, biochemical profiles on API 20 NE tests (BioMérieux, France) and in-house polymerase chain reaction (PCR) analysis (*Suppiah et al., 2010*). Prior to the study, the *B. pseudomallei* isolates were cultured aerobically in Luria Bertani (LB) broth at 37 °C and agitated at 180 rpm for 24 h according to *Al-Maleki et al. (2014)*.

### Peptide storage and handling

A total of 11 AMPs used in this study were selected based on the potential to inhibit other Gram-negative organisms; LL-37, Magainin 2, Tachyplesin 1 (TP1), PG1, Sushi 1, Sushi 2, 1037, 1018, DJK5, V2D, and ornithine. Among these LL-37, Magainin 2, TP1, and PG1 were synthesized by SBS Genetech, China; Sushi 1 and Sushi 2 were synthesised by First Base Laboratories, Singapore; Peptide 1037, 1018, and DJK5 were kindly provided by our collaborators from University of British Columbia, Canada; while V2D and ornithine were provided by our collaborators in National University of Singapore (Table 1).

The AMPs used in this study were stored in lyophilized form at −20 °C (*Travis et al., 1991*), and were dissolved with DNAase/RNAse-free distilled water (for all AMPs except sushi peptides), and 0.5% (v/v) of dimethyl sulfoxide (DMSO; for both Sushi peptides). Once dissolved, the AMPs were stored in glass vials in aliquots of 200 µl at

**Table 1** Sequences and characteristics of AMPs investigated.

| Peptides | Sequence | Relative molecular mass | Charge | Isolated from | References |
|---|---|---|---|---|---|
| LL-37 | LLGDFFRKSKEKIGKEFKRIVQRIKDFLRNLVPRTES | 4,493.33 | 6+ | Human neutrophils | *Gudmundsson et al. (1996)* |
| Magainin 2 | GIGKFLHSAKKFGKAFVGEIMNS | 2,466.9 | 4+ | *Xenopus laevis* | *Zasloff (1987)* |
| Tachyplesin 1 | KWCFRVCYRGICYRRCR | 2,263.7 | 6+ | *Tachypleus tridentatus* | *Miyata et al. (1989)* |
| Protegrin 1 | RGGRLCYCRRRFCVCVGR | 2,155.7 | 6+ | Porcine leukocytes | *Fahrner et al. (1996)* |
| Sushi 1 | GFKLKGMARISCLPNGQWSNFPPKCIRECA | 3,757 | 6+ | Horseshoe crab Factor C | *Yau et al. (2001)* |
| Sushi 3 | HAEHKVKIGVEQKYGQFPQGTEVTYTCSGNYFLM | 3,890 | 1+ | | |
| DJK 5 | Synthesized and provided by collaborator (University of British Columbia) | | | | |
| 1018 | | | | | |
| 1037 | | | | | |
| V2D | Synthesized and provided by collaborator (National University of Singapore) | | | | |
| Ornithine | | | | | |

a concentration of 10 mg/ml. The AMPs were diluted in 0.01% acetic acid containing 0.2% bovine serum albumin (BSA) for the preliminary screening whereas the dilution of AMPs with serum free Roswell Park Memorial Institute 1640 media (RPMI 1640; Life Technologies) was performed to minimize the dilution of the RPMI used in subsequent experiments. In addition, the microtiter plates used in this study were made of polypropylene while the storage vials were made of polypropylene (*Hancock, 1999*).

## Preliminary screening of AMPs antimicrobial activity on in *B. pseudomallei* isolates from melioidosis patients in Malaysia

The 100 strains mentioned were screened for AMP susceptibility by colony counting (*Sieuwerts et al., 2008*). The 11 AMPs were diluted to 1 mg/ml before use. Screening was performed as described previously (*Sheafor et al., 2008*) with slight modifications. Briefly, 80 µl of 24-h *B. pseudomallei* culture in LB broth was diluted with 1 X phosphate buffer saline (PBS; pH 7.4) to a bacteria density of $10^5$ CFU/ml and mixed with 20 µl AMPs prior to incubation. The mixture was then plated on fresh nutrient agar (NA) and incubated at 37 °C for 24 h to determine viability of the bacteria. Isolates were categorized as "sensitive" when no growth was observed and categorized as "resistant" if there one or more colonies grown after 24-h incubation on NA. Three biological replicates were performed.

## Minimum inhibitory concentration (MIC) and minimum bactericidal concentration (MBC) of AMPs on *B. pseudomallei* cells in planktonic state

Based on the preliminary screening results TP1 that inhibited *B. pseudomallei,* was selected for further study with LL-37 and PG1. In this study, *B. pseudomallei* strain K96243 and randomly selected *B. pseudomallei* clinical strains UMC031, UMC080, and UMC089, and *Escherichia coli* (ATCC 29522; control), were exposed to TP1, LL-37, and PG1 concentrations ranging from 0 to 200 µM, with two-fold increase. The exponential phase

culture of planktonic *B. pseudomallei* was centrifuged at 10,000 rpm for 15 min to pellet the bacteria. The resulting pellet was washed three times with PBS and the bacterial density was adjusted to a $10^5$ CFU/ml using serum free RPMI 1640 medium; as recommended by *Schwab et al. (1999)* in order to obtain the highest AMP activity. A total of 20 μl of each AMPs were added to 180 μl of the bacterial suspension in 96 well U-bottom microtiter plates (Eppendorf). Following incubation at 37 °C for 24 h, the plates were subjected to optical density (OD) measurement at 570 nm using a microplate absorbance reader (Tecan Sunrise, Männedorf, Switzerland) using the settings for U-bottom microtiter plates. Each well was subjected to a 10 times serial dilution and plated on NA to determine the viability of the bacteria (*Sieuwerts et al., 2008*). The MIC was read as the lowest concentration of AMP that inhibited visible growth of the bacteria (*Schwab et al., 1999*) compared to the untreated after 24-h incubation. Additionally, the MBC was determined when no growth was observed on NA following 24-h incubation. Three technical replicates were performed on three different occasions.

### Time-kill kinetic assay of *B. pseudomallei*

The time-kill kinetic assay was carried out with a similar set up as the MIC and MBC studies using *B. pseudomallei* strain K96243. Briefly, 20 μl of AMP (TP1 at 55, 110, and 221 μM; approximately the MIC and MBC of *B. pseudomallei* strains tested) was added to 180 μl of the bacterial suspension in a 96 well U-bottom microtiter plates. The plates were incubated 37 °C for 24 h and the absorbance was taken at OD 570 nm using the settings for U-bottom microtiter plates at 60-min interval. At every two hours, the bacterial suspension in each well was subjected to a 10 times serial dilution and plated on NA to determine bacteria viability (*Sieuwerts et al., 2008*). In addition, the inhibition patterns of TP1 was also compared with 54 μM of CAZ (30 μg/ml; equivalent to the concentration on the antibiotic disk; Oxoid, UK), and also the untreated bacteria. Three technical replicates were performed on three different occasions, and the data obtained were analyzed with one-way ANOVA followed by Dunnett's test to determine the significance relative to the untreated bacteria ($P < 0.05$).

### Inhibition activity of AMPs against *B. pseudomallei* in biofilm state

*B. pseudomallei* K96243 was grown using a modification of the Calgary Biofilm Device; (*Kanthawong et al., 2012*) also known as minimum biofilm eradication concentration (MBEC) assay (Innovotech, Edmonton, Canada). To test the inhibition activity in biofilm state, *B. pseudomallei* K96243 was exposed to 1 mg/ml, of the 11 tested AMPs including CAZ (30 μg/ml; 54 μM) and meropenem MRP (10 μg/ml; 26 μM). Both antibiotics were equivalent to the concentrations on the antibiotic disks (Oxoid, UK). In addition, TP1 at concentrations ranging from 55 to 2,649 μM (with two-fold increase) were also tested against *B. pseudomallei* K96243 biofilm. Briefly, 150 μl of *B. pseudomallei* culture (approximately $10^5$ CFU/ml) was inserted into individual wells of the MBEC assay plate before incubation at 37 °C for 16 h. Following incubation, the biofilms were rinsed into microtiter plates containing PBS and then transferred into a new micro titre plate

containing the antimicrobial agents tested in serum-free RPMI (antimicrobial challenge plates) before incubation at 37 °C for 24 h. Following these, the biofilms were disrupted using a water bath sonicator (Thermoline Scientific, Wetherill Park NSW, Australia) for 30 min in PBS and the viability of the bacteria from the biofilm (CFU/biofilm) was determined by colony counting (*Sieuwerts et al., 2008*). Three biological replicates were performed, and the data obtained were analysed with one-way ANOVA followed by Dunnett's test to determine the significance relative to the untreated bacteria ($P < 0.05$).

## Scanning electron microscopy (SEM)

SEM was performed according to *Song et al. (2012)* with slight modifications. Briefly, *B. pseudomallei* K96243 culture was incubated with 62 μM of TP1 at 37 °C for 24 h. Following centrifugation (10 min at $2,000 \times g$, 4 °C), the bacterial pellet was washed twice with PBS, and resuspended in 3.0% glutaraldehyde (incubated overnight at ambient temperature). Subsequently, the solution was rinsed three times with double distilled water and dehydrated in a graded series of ethanol solutions. After critical-point drying and layering with 20 nm gold coating (Leica EM SCD005, Leica Microsystems, Singapore), the microscopic analysis was performed using Quanta 650 FEG Scanning Electron Microscope (FEI, Oregon, USA). Both *B. pseudomallei* K96243 cell suspension without the AMPs and *E. coli* were treated as the control.

## Cytotoxicity of AMPs on mammalian cell lines

In this study, human lung epithelial (A549; ATCC® CCL185™), human gastric adenocarcinoma (AGS; ATCC® CRL-1739™) and human hepatocellular carcinoma (HepG2; ATCC® HB8065™) cell lines were cultured in RPMI 1640 containing foetal bovine serum (FBS) (10%) at 37 °C with 5% $CO_2$ and 95% relative humidity). Cytotoxicity was determined based on the reduction of 3-[4, 5-dimethylthiazol-2-yl]-2, 5 diphenyl tetrazolium bromide (MTT; Sigma, St. Louis, MO, USA) by cellular oxidoreductases of viable cells, which yields a crystalline blue formazan product (*Hansen et al., 2012*). Briefly, cells (approximately $2 \times 10^4$ cells/well) were seeded in tissue culture treated 96-well plates (Corning, Corning, NY, USA). After 24 h, the cells were rinsed with PBS and 200 μg/ml of the TP1 (110 μM), PG1 (115 μM) and LL-37 (55 μM) were added as a solution in fresh serum free medium to a final volume of 100 μl/well, respectively. Following 2 h of incubation, the MTT reagent (20 μl) was added to obtain a final concentration of 500 mg/l and further incubated for 3 h. After which DMSO (solubilizing solution) was added to lyse the cells and to dissolve the formazan crystals and the absorbance was recorded at OD 570 nm. The percentage of viable cells was calculated as followed:

$$\frac{(\text{Absorbance of peptide treated cells})}{(\text{Absorbance of untreated cells})} \times 100$$

Three technical replicates were performed on three different occasions, and the data obtained were analysed with one-way ANOVA followed by Dunnett's test to determine the significance relative to the untreated bacteria ($P < 0.05$). The AMPs effect on the cell lines were also compared to MRP (26 μM), and CAZ (54 μM).

## Possible TP1 interactions with protein targets from in silico molecular docking study

### Preparation of TP1 structure from PDB

Solution NMR structures of TP1 (PDB ID: 2RTV) were obtained from Protein Data Bank (PDB) (http://www.rcsb.org/pdb). The first model of 20 peptide configuration models available in the PDB file was used. Prior to docking, AutoDock tools 4.2.6 (ADT) software was used to prepare the ligand before proceeding to energy minimization using Accelrys Discovery Studio 2.5.5 (DS) software (Accelrys Software Inc., San Diego, CA, USA). The backbone of TP1 was kept rigid whereas most of the side chains were defined as flexible in molecular docking experiment.

### Interaction of TP1 on selected bacteria structures

A total of four bacteria structures were selected to visualize TP1 interaction; lipopolysaccharide (LPS) of *E. coli*, and *Streptococcus pneumoniae* proteins; autolysin, pneumolysin, and pneumococcal surface protein A (PspA).

TP1 has been reported to bind to the LPS of *E. coli* (*Kushibiki et al., 2014*). However, the LPS structure of *B. pseudomallei* is yet to be reported. Thus, the LPS model from *E. coli* was used to predict the interaction of TP1 with *B. pseudomallei* as both are Gram-negative bacteria and *E. coli* is often used as a representative for Gram negative bacteria (*Fernandez-Recio et al., 2004*; *Galdiero et al., 2012*). The molecular docking study was carried out with the prepared TP1 structure and the LPS structure of *E. coli* (PDB ID: 1QFG) based on *Kushibiki et al. (2014)* with a slight modification to add the missing atoms (e.g. hydrogens) in the model. The initial structure was modified according to the CHARMm force field with partial charge Momany-Rone (*Momany & Rone, 1992*) and minimizations of the structures were performed, satisfying the root mean squared gradient tolerance of 0.1000 kcal/(mol × Angstrom) before docking was carried out using AutodockVina (ADV) (*Trott & Olson, 2010*) instead of Autodock 4.2 as published. A 70 × 80 × 80-point grid box of the structure was then generated with a grid spacing of 0.375 Å, and centred on GlcN II in lipid A between TP1 and the LPS structure.

TP1 was reported to bind to both Gram negative and Gram positive bacteria (*Imura et al., 2007*; *Ohta et al., 1992*) but to date, the binding activity on S. *pneumoniae* proteins was yet to be documented. As previously reported by *Le et al. (2015)*, three possible *S. pneumoniae* proteins targets for antimicrobial activities were autolysin, pneumolysin, and PspA (virulence factors). Therefore, TP1 was docked on three S. *pneumoniae* protein structures; autolysin (PDB ID: 1GVM), pneumolysin and PspA according to *Le et al. (2015)*. The binding sites for the three molecules were defined as; autolysin at chain B: LYS258-ALA277, for pneumolysin at ARG426-ARG437, for PspA at GLY577-LEU588. The minimizations of these structures were carried out with a similar protocol as LPS of *E. coli*. A 40 × 40 × 40-point grid box of the structure was then generated with a grid spacing of 0.375 Å, and centred at the respective binding sites. For all four structures, the interaction energies (IE; the summation of van der Waals' (VDW) and Electrostatic IE) were investigated between TP1 and the structures using DS software. A low (negative) interaction energy between TP1 and the bacterial structures indicates a

stable system and thus likely to bind. In addition, the software was also used to detect intermolecular hydrogen ($H^+$) bonds and hydrophobic interactions which play a role in stabilizing the docked complex.

### Interaction of TP1 on homology modelled B. pseudomallei protein structure

At this stage, all the three S. *pneumoniae* proteins demonstrated negative IE with TP1 which indicated a strong interaction. When we searched for the protein sequences (BLAST search; blastp; non-redundant proteins with 95% sequence similarity) for all the three protein structures against *B. pseudomallei* protein sequence database, only the PspA sequence secured a hit in the search; YD repeat-containing protein (accession: CFU00865). This may indicate that there are high similarities in the proteins of the Gram positive and Gram negative bacteria. Similarly, *Alloing, Trombe & Claverys (1990)* have also demonstrated that the Ami proteins of *S. pneumoniae* exhibited homology with components of the oligopeptide permeases of the Gram-negative *Salmonella typhimurium* and *E. coli*. In addition, autolysin exists in the peptidoglycan bacterial cell walls, which applies to both Gram positive and Gram negative bacteria (*Beveridge, 1999*). Thus, it is possible to execute homology modelling using the protein sequences of either Gram-positive or Gram-negative due to the similarities between both groups of bacteria. Taking into consideration of the above points, we have carried out homology modelling of the *B. pseudomallei* protein using PspA as a template.

Using DS, multiple sequence alignment was performed and the model was built using MODELLER function. The overall quality of the minimized model was evaluated by utilizing PROCHECK (*Laskowski et al., 1993*) for evaluation of Ramachandran plot quality, ERRAT (*Colovos & Yeates, 1993*) to verify patterns of non-bonded atomic interactions, and VERIFY3D (*Lüthy, Bowie & Eisenberg, 1992*) for assessing the compatibility of each amino acid residue. Subsequently with ADV, TP1 was docked onto the YDP model using the binding site of PspA as a reference in order to predict the intermolecular interaction.

## RESULTS

### Preliminary screening of AMPs antimicrobial activity on B. pseudomallei isolates from Malaysia

A total of 100 *B. pseudomallei* isolates including the reference strain *B. pseudomallei* K96243 was subjected to colony counting after the exposure to the AMPs. Eighty-three strains (83%) were susceptible to TP1 whereas 100 strains (100%) were susceptible to both PG1 and LL-37. The isolates were 100% resistant to the remainder AMPs. In addition, there was no correlation between the AMP activity profiles and the antibiotic susceptibility data available (Supplemental Information 1 and 2).

Based on the observation, although TP1 did not completely inhibit all the tested *B. pseudomallei* isolates as seen with both LL-37 and PG1, it still was able to inhibit the isolates compared to the remainder eight AMPs which did not inhibit the tested isolates. Here, TP1 demonstrated the highest inhibition potential, comparable to LL-37

**Table 2 Summary of MIC and MBC values of PG1, LL-37 and TP1.**

| AMP tested | *B. pseudomallei* (average) | | *E.coli* (ATCC 29522) | |
| --- | --- | --- | --- | --- |
| | MIC (μM) | MBC (μM) | MIC (μM) | MBC (μM) |
| Protegrin 1 (Control) | 15.46 ± 5.02 | 15.46 ± 6.70 | 5.8 ± 1.09E-15 | 14.24 ± 2.14 |
| LL-37 (Control) | 47.29 ± 8.35 | 92.73 ± 8.03 | 14.8 ± 6.42 | 14.8 ± 6.42 |
| Tachyplesin 1 | 61.69 ± 6.38 | 193.35 ± 2.00E-14 | 22.1 ± 0 | 22.1 ± 0 |

and PG1. To date, *B. pseudomallei* susceptibility to the eight non-responsive AMPs is yet to be reported whereas there were reports of *B. pseudomallei* susceptibility to LL-37 (*Kanthawong et al., 2012*), and PG1 (*Sim et al., 2011*). As such, TP1 was selected for further investigations in order to suggest its efficacy in comparison with both LL-37 and PG1.

## MIC and MBC on planktonic *B. pseudomallei* cells

The MIC of TP1 (61.69 μM) was threefold lower than its MBC (193.35 μM). Relative to the MIC of TP1, MIC levels of LL37 (47.29 μM) and PG1 (15.46 μM) were approximately 1.3 and four-fold lower, respectively (Table 2). Similarly, the MBC of TP1 demonstrated two-fold and 12-fold higher levels compared to LL37 (92.73 μM) and PG1 (15.46 μM), respectively. Overall, both the MIC and MBC levels of TP1 were found to be higher than that of LL-37 and PG1. On the other hand, the MIC and MBC values of *B. pseudomallei* for all three AMPs were observed higher than *E. coli*. In conclusion, TP1 exhibited *B. pseudomallei* inhibition, although not as potent as PG1 and LL-37.

## Time-kill kinetic assay of *B. pseudomallei*

At 221 μM of TP1, the growth inhibition of *B. pseudomallei* K96243 was observed within two hours after exposure compared to that of CAZ (8 h) (Fig. 1). At 110 μM, TP1 managed to inhibit the growth of *B. pseudomallei* K96243 following eight hours of exposure compared to the growth of the untreated bacteria at the same time point (approximately 9.0 log CFU/ml), similar to CAZ. At 55 μM, there was a slight inhibition of the growth of *B. pseudomallei* K96243 (9.96 log CFU/ml) as compared to the untreated bacteria (10.17 log CFU/ml) at 24 h, albeit not significant. Overall, TP1 was able to inhibit *B. pseudomallei* K96243 growth at a concentration above at 55 μM and in a shorter duration as compared to CAZ.

## Inhibition activity of AMPs against *B. pseudomallei* in biofilm state

*B. pseudomallei* K96423 in biofilm state was exposed to the 11 different AMPs (1 mg/ml) using MBEC assay (Fig. 2). Of the AMPs tested, only TP1, 1018, and PG1 demonstrated significant inhibition (4.2 log CFU/biofilm, 7.2 log CFU/biofilm, and 1.1 log CFU/biofilm, respectively) compared to the untreated bacteria (7.8 log CFU/biofilm). CAZ and MRP demonstrated complete inhibition. At the same, using 1 mg/ml of TP1, LL-37, and PG1 (which was equivalent to 442, 222, and 464 μM, respectively), the inhibition activity of TP1 was 3.5-log lower than PG1 but approximately 2.0-log higher
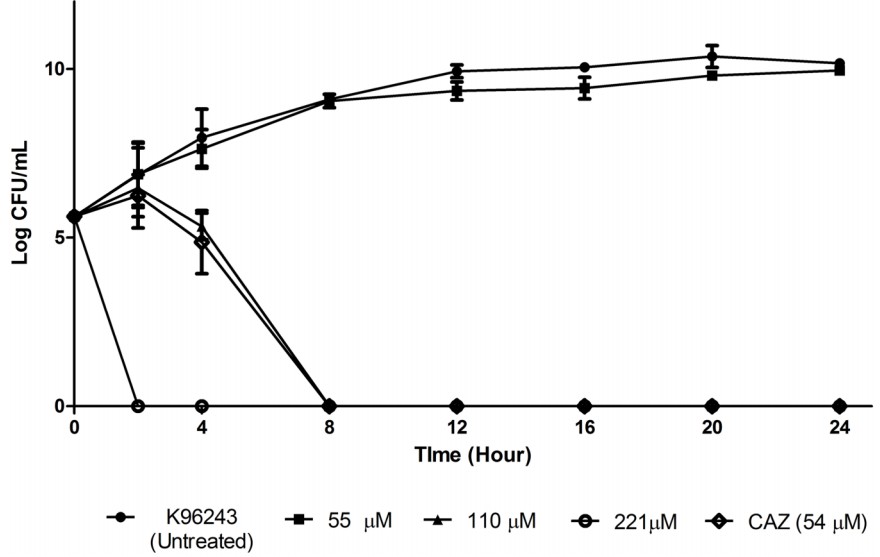

**Figure 1 Time killing curve of TP1 compared to that of CAZ for *B. pseudomallei* K96243.** *B. pseudomallei* K96243 was exposed to TP1 (55, 110, and 221 $\mu$M) and CAZ (54 $\mu$M). Data was presented as mean and standard deviation of three independent replicates, analyzed with one-way ANOVA followed by Dunnett's test to determine the significance relative to the untreated bacteria (P < 0.05).

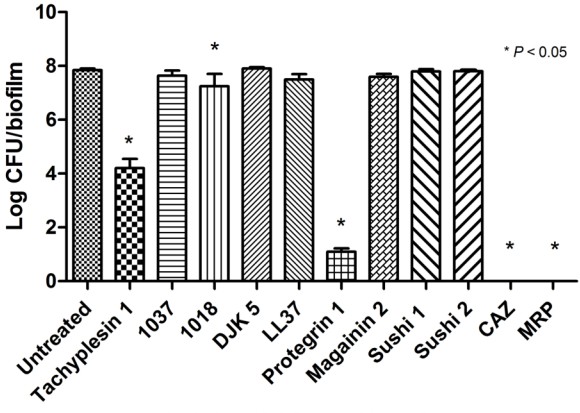

**Figure 2 Number of surviving *B. pseudomallei* K96243 in biofilm state post 24-h post antimicrobial peptides exposure.** The concentration of antimicrobial agents used were 1 mg/ml. TP1, 1018, PG1, CAZ and MRP showed significant reduction compared to the untreated. These experiments were conducted in three independent replicates. The error bars indicate the standard deviation. One-way ANOVA followed by Dunnett's test was performed to determine the significance relative to the untreated bacteria (P < 0.05; indicated by *).

than LL-37 compared to the untreated bacteria (7.8 log CFU/biofilm). There was no significant inhibition between LL-37-treated bacteria and the untreated bacteria. The inhibition activity of TP1 was higher that LL-37, but lower than CAZ, MRP, and PG1.

When *B. pseudomallei* K96243 in biofilm state was exposed to increasing concentrations of TP1 (from 55 to 2,649 $\mu$M; two-fold increase), the number of cells gradually decreased from 7.84 log CFU/biofilm to 3.6 log CFU/biofilm (Fig. 3). There was a significant decrease in the number of cells when exposed to all the TP1

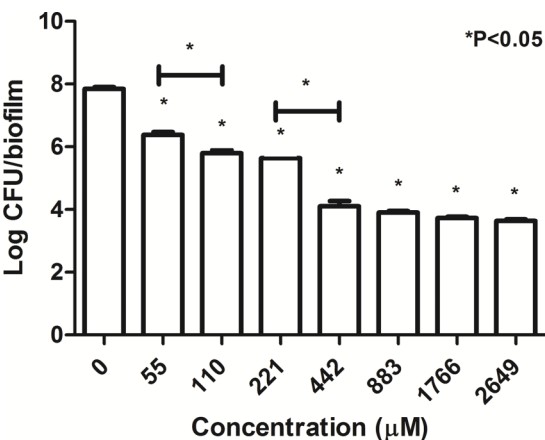

**Figure 3** Number of surviving *B. pseudomallei* K96243 in a biofilm state 24-h post-exposure to TP 1. *B. pseudomallei* K96243 was exposed to different concentrations of TP1 (55–2649 μM, with twofold increase). These experiments were performed in three independent replicates where the error bars indicate the standard deviation. Data was analyzed with one-way ANOVA followed by Dunnett's test to determine the significance relative to the untreated bacteria (P < 0.05; indicated by *).

concentrations as compared to the untreated cells (0 μM). In addition, TP1 was observed to suppress the growth of the cells in biofilm state at concentrations above 442 μM (~4.00 log CFU/biofilm; ~2.0-log CFU/biofilm reduction compared to the untreated) although MBEC was not achieved. There were significant growth inhibitions observed when the bacteria were exposed to TP1 at 55 and 100 μM (1.0-log reduction) as well as 221 and 442 μM (1.3-log reduction). In general, TP1 was able to reduce the number of *B. pseudomallei* in biofilm state from concentrations above 55 μM.

## Scanning electron microscopy (SEM)

The untreated *B. pseudomallei* cells, in basal RPMI medium, displayed a smooth and intact surface (Fig. 4A). When the cells were exposed to TP1, some of the cells looked corrugated with dimples on the surface, and the appearance of blisters and bubbles were also observed on the membrane (Fig. 4B). Moreover, some of the cells were also observed to lose their original structure, leaving cell debris intermixed with the cell membranes that were blebbing. Similar observation was also seen with *E. coli* (Figs. 4C and 4D).

## Cytotoxicity of AMPs on mammalian cell lines

As the concentration of TP1 increased (from 2.7 to 110.4 μM), the percentage of viable cells for all three cell lines used were found to be decreased (from 92.0 to 15%, respectively for A549, 94.1 to 8.3%, respectively for HEPG2, and 85.1 to 50.1%, respectively for AGS). Besides that, at 48.6 and 110.4 μM of TP1, there was a significant reduction of cell viability in all three cell lines compared to that of lower concentrations of TP1 (from 2.7 to 22.3 μM). Among the three cell lines tested, at 48.6 and 110.4 μM, AGS showed a higher percentage of viability (80 and 50%, respectively) compared to A549 (57 and 15%, respectively) and HEPG2 (57 and 8%, respectively). On the other hand, at lower concentrations of TP1 from 2.7 to 22.3 μM, HEPG2 showed higher percentage of viability (90.5 ± 7.6%) compared to that of AGS (90.0 ± 8.4%) and

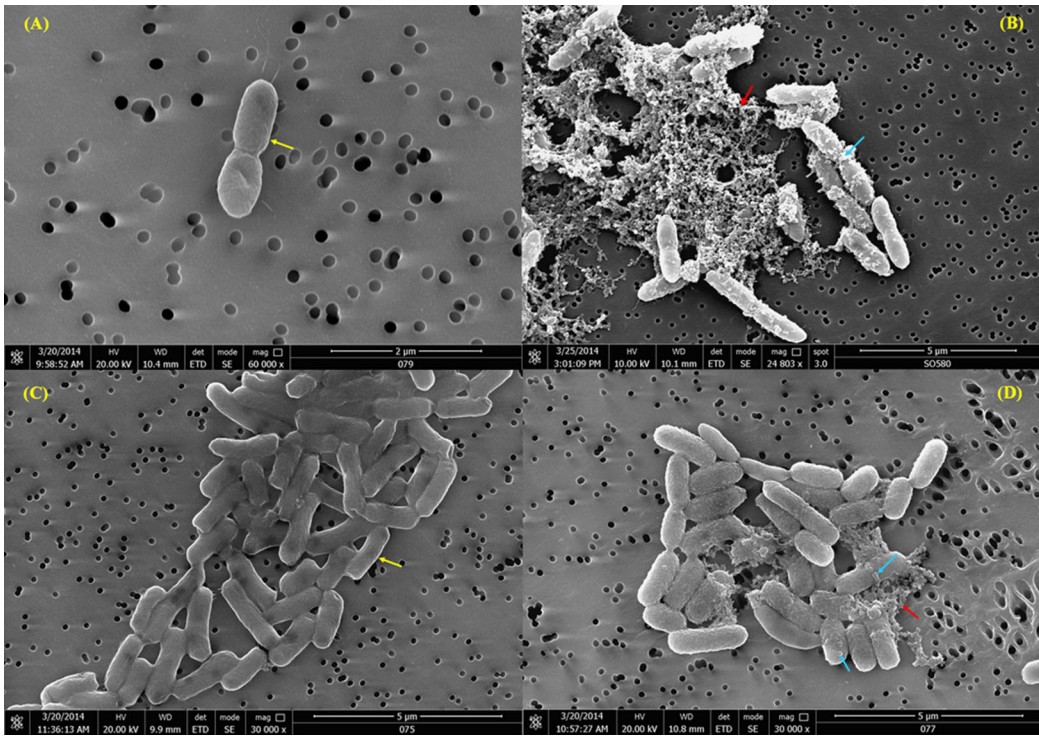

**Figure 4 (A) SEM observation of *B. pseudomallei* K96243 pre-exposure; (B) post-exposure of 62 μM TP1; (C) *E. coli* ATCC29522 pre-exposure and (D) post-exposure of 22 μM of TP1.** The bacteria samples were fixed on a membrane attached to a double-sided adhesive tape on an SEM stub. Yellow arrows indicate bacterial cell membrane structure before exposure to TP1, blue arrows indicate membrane blebbing, and red arrows indicate cell debris.

A549 (85.8 ± 9.6%). When the same amount of peptides used (200 μg/ml which is equivalent to 110.4 μM of TP1 and 115 μM of PG1), TP1 showed lesser cytotoxicity (15.0% for A549, 8.3% for HEPG2, and 50.1% for AGS) compared to that of PG1 (3.3% for A549, 5.0% for HEPG2, and 3.0% for AGS) (Fig. 5). Overall, HEPG2 cells demonstrated higher percentage of cell viability compared to the AGS at the lower concentrations of TP1 (2.7–11.0 μM), albeit at a not significant level. However, AGS demonstrated significantly higher percentage of cell viability ($P < 0.05$) compared to HEPG2 at the higher concentrations of TP1 (48.6–110.4 μM). In general, TP1 has lower cytotoxicity at concentrations between 2.7 to 22.3 μM where HEPG2 demonstrated more tolerance to TP1.

At this stage, TP1 demonstrated inhibition activities on the tested *B. pseudomallei* strains comparable to LL-37 and PG1. SEM also revealed that TP1 was able to induce membrane blebbing, disrupting the membrane integrity and eventually leading to cell death. To date, the exact target on *B. pseudomallei* for TP1 binding has yet to be reported but there were existing reports of TP1 binding to the *LPS* (*Kushibiki et al., 2014*) and membrane (*Hong et al., 2015*) of *E. coli* as well as the minor groove of the DNA (*Yonezawa et al., 1992*). Therefore, in silico molecular docking was carried out to predict the binding targets of TP1 on *B. pseudomallei*.

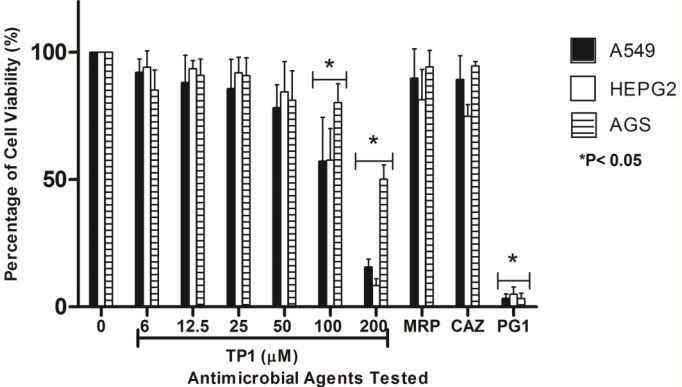

**Figure 5  Percentage of viability of A549, HEPG2 and AGS cell lines post-exposure to TP1, PG1, MRP and CAZ.** The cell lines were exposed to TP1 (0, 2.7, 5.5, 11.0, 22.3, 48.6, and 110.4 μM), PG1 (115 μM), MRP (26 μM), and CAZ (54 μM). These experiments were conducted in three independent replicates where the error bars indicate the standard deviation. Data was analysed with one-way ANOVA followed by Dunnett's test to determine the significance relative to the untreated bacteria (P < 0.05; indicated by *).

## Possible TP1 interactions with protein targets from in silico molecular docking study

### Interaction of TP1 to the LPS of E. coli

TP1 was observed to interact with the LPS of *E. coli* with a binding affinity from ADV of −5.4 kcal/mol and an IE of −838.34 kcal/mol where the interaction was mostly contributed by electrostatic −810.79 kcal/mol compared to VDW (−27.55 kcal/mol; Table 3). The IE of TP1 was mostly contributed by residues LYS1 and ARG 15 (−121.19 and −239.74 kcal/mol, respectively) while the IE of LPS was contributed mostly by the PO4 residue (−356.16 kcal/mol). Moreover, nine hydrogen bonds were formed between TP1 and the LPS of *E. coli* at TP1:LYS1:HT1–LPS: FTT1010:O3, TP1:ARG15: HH11–LPS: KDO1002:O1A, TP1:ARG15: HH12–LPS: PO42001:O1, TP1:ARG15: HH12–LPS: PO42001:O2, TP1:ARG15: HH22–LPS: GCN1001:O4, TP1:ARG15: HH22–LPS: PO42001:O1, TP1:ARG17: HE–LPS: KDO1003:O7, TP1:ARG17: HE–LPS: KDO1003:O8, and TP1:ARG17: HH22–LPS: KDO1003:O8, where the hydrogen bonds were formed mostly at the terminal ends of TP1 (Fig. 6). However, there were no hydrophobic interactions observed between the TP1 and LPS. As such, the TP1-LPS docked complex was stabilized the intermolecular hydrogen bonds formed.

### Interaction of TP1 to Streptococcus pneumoniae protein structures

Based on the molecular docking using ADV, TP1 interacted with all three *S. pneumoniae* proteins with a binding affinity of −8.1 kcal/mol for autolysin, −7.7 kcal/mol for pneumolysin, and −7.3 kcal/mol for PspA (Table 4). This indicates that TP1 has a higher binding affinity to autolysin, followed by pneumolysin, and PspA. In addition, the number of hydrogen bonds formed between TP1 and the *S. pneumoniae* proteins demonstrated a similar trend to the binding affinity where the autolysin (with the highest binding affinity) formed 13 bonds with TP1, followed by pneumolysin (nine bonds) and PspA (eight bonds). However, based on the IE, TP1 demonstrated stronger interaction

**Table 3 Contribution of the interactions energy in kcal/mol of the *E. coli* LPS binding residues with TP1.**

| Residue | Interaction energy (kcal/mol) | VDW interaction energy (kcal/mol) | Electrostatic interaction energy (kcal/mol) | Residue | Interaction energy (kcal/mol) | VDW interaction energy (kcal/mol) | Electrostatic interaction energy (kcal/mol) |
|---|---|---|---|---|---|---|---|
| TP1_LYS1 | −121.19 | −7.95 | −113.24 | 1QFG_PA11000 | −26.62 | −0.15 | −26.47 |
| TP1_TRP2 | −29.60 | −5.32 | −24.29 | 1QFG_GCN1001 | −52.60 | −2.51 | −50.09 |
| TP1_CYS3 | −23.35 | −0.05 | −23.29 | 1QFG_KDO1002 | −44.60 | −5.90 | −38.70 |
| TP1_PHE4 | −25.45 | −5.59 | −19.86 | 1QFG_KDO1003 | −37.97 | −1.86 | −36.11 |
| TP1_ARG5 | −32.65 | −0.19 | −32.46 | 1QFG_GMH1004 | −6.17 | −0.17 | −6.00 |
| TP1_VAL6 | −13.70 | −0.52 | −13.18 | 1QFG_GMH1005 | −32.40 | −1.18 | −31.22 |
| TP1_CYS7 | −13.12 | −0.01 | −13.11 | 1QFG_GLC1006 | −23.80 | −0.35 | −23.45 |
| TP1_TYR8 | −15.07 | 0.00 | −15.06 | 1QFG_GLC1007 | −34.08 | −2.72 | −31.36 |
| TP1_ARG9 | −29.86 | −0.01 | −29.85 | 1QFG_GLA1008 | −1.37 | 0.00 | −1.36 |
| TP1_GLY10 | −26.80 | −0.06 | −26.74 | 1QFG_FTT1009 | −28.99 | −2.56 | −26.43 |
| TP1_ILE11 | −33.80 | −0.51 | −33.29 | 1QFG_FTT1010 | −23.81 | −3.59 | −20.23 |
| TP1_CYS12 | −27.53 | −0.16 | −27.37 | 1QFG_FTT1011 | −24.64 | −2.24 | −22.41 |
| TP1_TYR13 | −47.91 | −7.22 | −40.69 | 1QFG_DAO1012 | −0.16 | −2.61 | 2.45 |
| TP1_ARG14 | −44.16 | −0.45 | −43.71 | 1QFG_FTT1013 | −27.94 | −5.13 | −22.81 |
| TP1_ARG15 | −239.74 | 4.28 | −244.01 | 1QFG_MYR1014 | −35.22 | −4.19 | −31.03 |
| TP1_CYS16 | −28.67 | −0.21 | −28.46 | 1QFG_NI1030 | −48.15 | 0.00 | −48.15 |
| TP1_ARG17 | −85.75 | −3.58 | −82.17 | 1QFG_PO42001 | −356.16 | 7.64 | −363.80 |
|  |  |  |  | 1QFG_PO42005 | −33.66 | −0.04 | −33.62 |
| Total | −838.34 | −27.55 | −810.79 |  | −838.34 | −27.55 | −810.79 |

**Note:**
Details of LPS abbreviation are available in Supplemental Information 4.

with PspA (−573.48 kcal/mol), followed by autolysin (−444.14 kcal/mol), and pneumolysin (−135.59 kcal/mol). Moreover, there were additional intermolecular interactions between TP1-penumolysin and TP1-PspA docked complexes but none was detected with TP1-autolysin docked complex. TP1-pneumolysin docked complex was observed to contain two pi-cation interactions between ARG9 of TP1 to TRP433 of pneumolysin. On the other hand, TP1-PspA docked complex demonstrated three types of additional interactions; one pi-pi interaction (TRP2 to TYR 541 of PspA), four pi-cation interactions (LYS1 to TRP 525 of PspA; three bonds from LYS1 to TRP518 molecules), and a pi-sigma interaction between LYS1 to TRP 518 of PspA. Based on the data obtained, among the three *S. pneumoniae* proteins, TP1 was predicted to bind to autolysin at close proximity but was predicted to form the strongest interaction and the most stable docked complex with PspA.

### Interaction of TP1 on homology modelled *B. pseudomallei* protein structure

Homology model of PspA *B. pseudomallei* (YDP model) was evaluated to ensure acceptable model quality (Supplemental Information 5). TP1 was observed to interact with the YDP model with a binding affinity of −7.6 kcal/mol and an IE of −822.80 kcal/mol where the interaction was mostly contributed by electrostatic

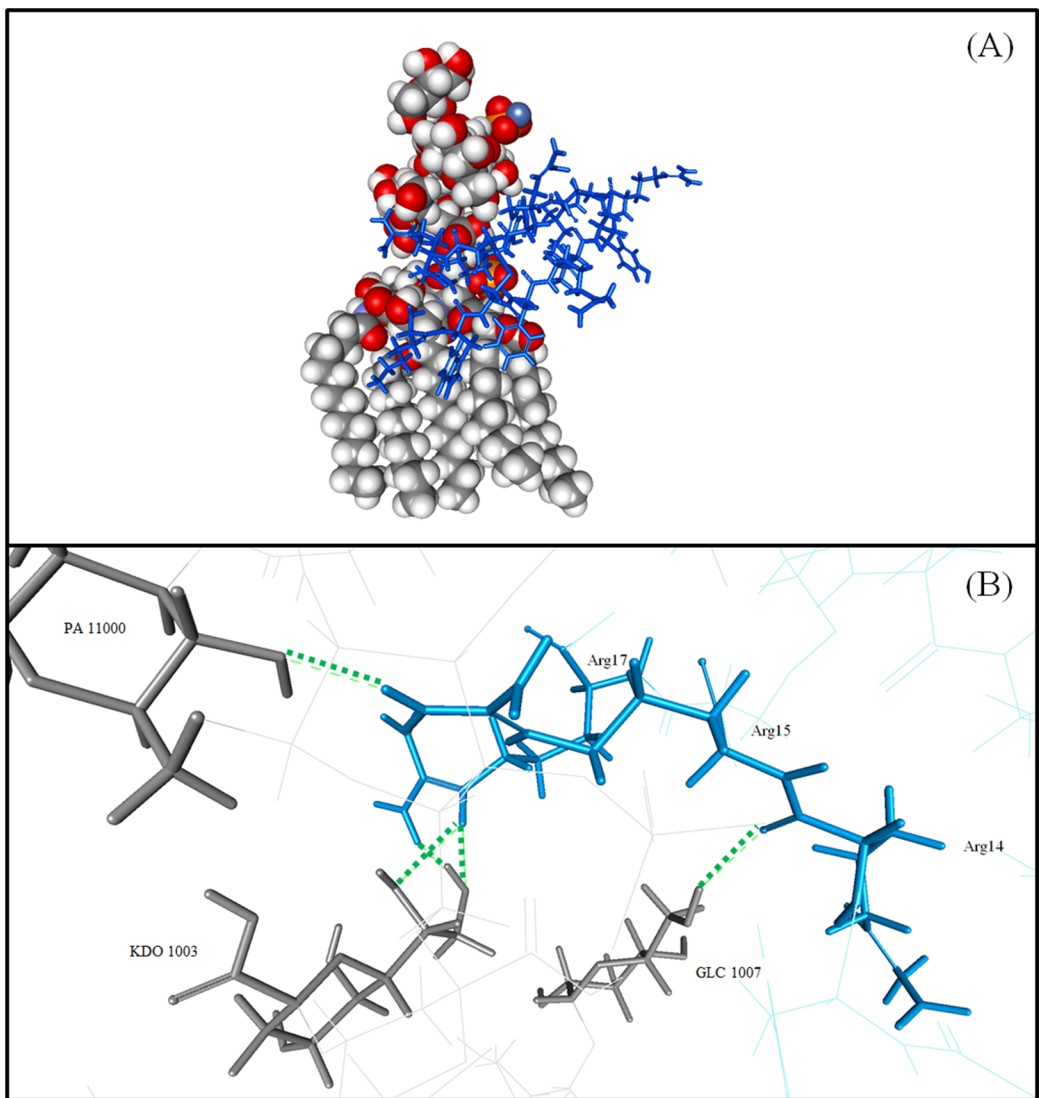

**Figure 6 Graphical representation of LPS of *E. coli* and TP1 interaction.** (A) Superimposed TP1 was coloured blue while the LPS molecule was coloured grey. (B) The binding amino acids and hydrogen bonds were indicated by green dashes. Five hydrogen bonds were formed, mostly at the terminal ends of TP1:ARG15: HH22–A: PA11000:O4, TP1:ARG17: HE–A: KDO1003:O7, TP1:ARG17: HE–A: KDO1003: O8, TP1:ARG17: HH22–A: KDO1003:O8, and A: GLC1007:H6–TP1:ARG14: O.

IE −770.63 kcal/mol compared to VDW IE (−52.17 kcal/mol; Table 5). The binding affinity and the IE indicated that TP1 will bind to the YDP model in close proximity with a strong intermolecular interaction. The IE of TP1 was mostly contributed by residues LYS1, ARG 14, and ARG 15 (−158.78, −160.54, and −124.82 kcal/mol, respectively) while the IE of YDP was contributed mostly by ASP 1220 (−209.81 kcal/mol) and ASP 1169 (−190.00 kcal/mol). Moreover, nine hydrogen bonds were observed between TP1 and YDP model at TP1:LYS1: HZ2 to YDP: THR1113: O, TP1:CYS12:HN to YDP: SER1185: OG, TP1:ARG14: HE to YDP: ASP1169:OD1, TP1:ARG14: HE to YDP: ASP1169:OD2, TP1:ARG14: HH22 to YDP: ASP1169:OD1, TP1:ARG17: HH22 to YDP:

**Table 4 Summary of TP1 interaction with S. pneumoniae protein structures.**

| Docked complex | Binding affinity (kcal/mol) | Interaction energy (kcal/mol) | VDW interaction energy (kcal/mol) | Electrostatic interaction energy (kcal/mol) | No. of hydrogen bonds | Hydrogen bond details | Additional intermolecular interactions |
|---|---|---|---|---|---|---|---|
| TP1-autolysin (B) | −8.1 (−7.6) | −444.17 | −51.03 | −393.14 | 13 | TP1:LYS1:HT1–B:ASP246: O<br>TP1:LYS1:HT2–B: ASP246:OD2<br>TP1:LYS1:HT2–B:ASP246: O<br>TP1:LYS1:HT3–B: ASP246:OD2<br>TP1:LYS1: HZ1–B: VAL279: O<br>TP1:LYS1: HZ2–B: MET278: O<br>TP1:LYS1: HZ3–B: SER280: OG<br>TP1:TRP2: HE1–B: SER280: OG<br>TP1:ARG15: HH11–B: SER280: O<br>TP1:ARG15: HH12–B: ASN281:OD1<br>TP1:ARG15: HH22–B: ASN281:OD1<br>B: LYS274:HZ2–TP1:ARG5: O<br>B: LYS274:HZ3–TP1:ARG5: O | None |
| TP1-pneumolysin (P) | −7.7 (−6.2) | −135.59 | −37.62 | −97.97 | 9 | TP1:ARG14: HH22–P: GLU434:OE2<br>TP1:ARG15: HE–P: ASN400:OD1<br>TP1:ARG15: HH21–P: ASN400:OD1<br>TP1:ARG15: HH21–P: ASN400: O<br>TP1:ARG17: HH12–P: ASP380:OD2<br>TP1:ARG17: HH21–P: THR378:OG1<br>TP1:ARG17: HH22–P: THR378:OG1<br>P: LYS424:HZ1–TP1:ARG17: O<br>P: GLU434:HN–TP1:GLY10: O | Pi-Cation:<br>P: TRP433–TP1:ARG9:NE<br>P: TRP433–TP1:ARG9:NE |
| TP1-pspA | −7.3 (−6.4) | −573.48 | −46.10 | −527.38 | 8 | TP1:LYS1: HZ2–pspA: ASN571: O<br>TP1:TRP2: HE1–pspA: TYR546: OH<br>TP1:GLY10:HN–pspA: LYS580: O<br>TP1:TYR13: HH–pspA: ASP573: OD2<br>TP1:ARG15: HH21–pspA: ASN542: O<br>TP1:ARG15: HH21–pspA: ASN542: O<br>TP1:ARG17: HH22–pspA: ASN542: O<br>pspA: ASN569:HD21–TP1:TYR13: OH | Pi-Pi:<br>PspA: TYR541–TP1:TRP2<br><br>Pi-Cation:<br>pspA: TRP525–TP1:LYS1:NZ<br>pspA: TRP518–TP1:LYS1: N<br>pspA: TRP518–TP1:LYS1: N<br>pspA: TRP518–TP1:LYS1:NZ<br><br>Pi-Sigma:<br>pspA: TRP518–TP1:LYS1: HB1 |

**Table 5 Contribution of the interactions energy in kcal/mol of the homology modelled *B. pseudomallei* protein (YDP) binding residues with TP1.**

| Residue | Interaction energy (kcal/mol) | VDW interaction energy (kcal/mol) | Electrostatic interaction energy (kcal/mol) | Residue | Interaction energy (kcal/mol) | VDW interaction energy (kcal/mol) | Electrostatic interaction energy (kcal/mol) |
|---|---|---|---|---|---|---|---|
| TP1_LYS1 | −158.78 | −6.00 | −152.78 | YDP MODEL_THR1113 | −26.00 | −1.90 | −24.10 |
| TP1_TRP2 | −39.15 | −4.41 | −34.74 | YDP MODEL_GLY1114 | −9.83 | −0.89 | −8.94 |
| TP1_CYS3 | −22.10 | −0.41 | −21.69 | YDP MODEL_ASN1115 | −5.55 | −2.25 | −3.31 |
| TP1_PHE4 | −15.84 | −0.22 | −15.62 | YDP MODEL_ARG1119 | 115.99 | −7.04 | 123.03 |
| TP1_ARG5 | −43.61 | −0.24 | −43.37 | YDP MODEL_ASP1120 | −209.81 | −0.04 | −209.77 |
| TP1_VAL6 | −1.99 | −0.12 | −1.87 | YDP MODEL_MET1121 | −22.28 | −0.85 | −21.43 |
| TP1_CYS7 | −16.13 | −0.66 | −15.47 | YDP MODEL_ASN1152 | −13.83 | −0.39 | −13.44 |
| TP1_TYR8 | −3.87 | −0.64 | −3.23 | YDP MODEL_LEU1153 | −15.54 | −1.94 | −13.59 |
| TP1_ARG9 | −37.61 | −1.29 | −36.32 | YDP MODEL_SER1155 | −18.76 | −2.36 | −16.39 |
| TP1_GLY10 | −19.93 | −1.69 | −18.24 | YDP MODEL_ALA1156 | −9.01 | −1.78 | −7.23 |
| TP1_ILE11 | −6.20 | −2.93 | −3.27 | YDP MODEL_ARG1157 | 3.56 | −6.45 | 10.01 |
| TP1_CYS12 | −19.07 | −5.30 | −13.77 | YDP MODEL_TYR1166 | −13.50 | −2.83 | −10.67 |
| TP1_TYR13 | −56.36 | −4.94 | −51.42 | YDP MODEL_GLY1167 | −12.56 | −0.52 | −12.04 |
| TP1_ARG14 | −160.54 | −11.89 | −148.65 | YDP MODEL_TYR1168 | −13.56 | −3.68 | −9.88 |
| TP1_ARG15 | −67.65 | −1.55 | −66.10 | YDP MODEL_ASP1169 | −190.00 | −5.01 | −184.99 |
| TP1_CYS16 | −29.17 | −3.82 | −25.35 | YDP MODEL_ASN1171 | −14.29 | −0.88 | −13.41 |
| TP1_ARG17 | −124.82 | −6.07 | −118.74 | YDP MODEL_LEU1177 | 2.96 | −0.34 | 3.30 |
| | | | | YDP MODEL_ASP1179 | −81.10 | −2.25 | −78.86 |
| | | | | YDP MODEL_PRO1180 | −20.33 | −3.95 | −16.39 |
| | | | | YDP MODEL_SER1185 | −21.11 | −4.17 | −16.94 |
| | | | | YDP MODEL_ARG1223 | −12.49 | −0.41 | −12.09 |
| Total | −822.80 | −52.17 | −770.63 | | −587.05 | −49.93 | −537.12 |

ASP1120:OD2, YDP: SER1155: HG to TP1:CYS16: O, YDP: ARG1157:HH12–TP1:CYS12: O, and YDP: TYR1166: HH to TP1:CYS12: O (Fig. 7). However, there were no hydrophobic interactions observed between the TP1 and YDP. In general, the TP1-YDP model was stabilized by the intermolecular hydrogen bonds formed.

Along with the common binding protein targets for peptide and inhibitors, the PDB database search was performed and the available *B. pseudomallei* PDB structures (Supplemental Information 6) were subjected to molecular docking to predict the possible interactions with TP1 (Supplemental Information 7). Of the 26 *B. pseudomallei* proteins retrieved PDB (including stress proteins, secretion needle proteins, and penicillin binding pump) binding affinities from −5.7 to −11.2 kcal/mol were observed. Besides that, seven proteins with the interaction of amino acids in target receptor from peptide less than −100 kcal/mol (with CHARMm force field) were then further analysed to identify additional hydrogen bonds and intermolecular pi-interactions (Supplemental Information 8). TP1 demonstrated the most negative interaction energy with the cell inhibiting factor (PDB ID: 3GQM; −230.75 kcal/mol) with a binding

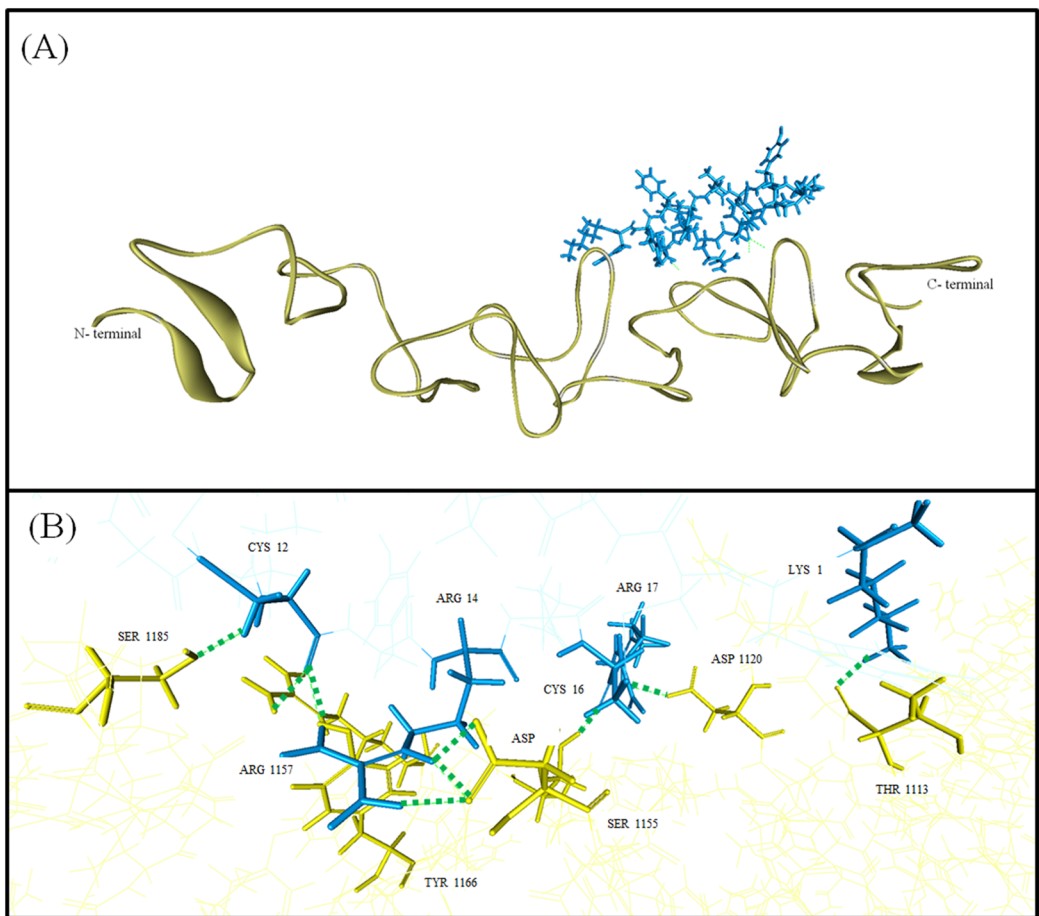

**Figure 7 Graphical representation of YDP model and TP1 interaction.** (A) Superimposed TP1 was coloured blue while YDP model was coloured yellow. (B) The binding amino acids and hydrogen bonds were indicated with green dashes. Nine hydrogen bonds were observed between TP1 and YDP: TP1:LYS1 to YDP: THR1113, TP1:CYS12 to YDP: SER1185, TP1:ARG14 to YDP: ASP1169, TP1:ARG14 to YDP: ASP1169, TP1:ARG14 to YDP: ASP1169, TP1:ARG17 to YDP: ASP1120, YDP: SER1155 to TP1:CYS16, YDP: ARG1157 to TP1:CYS12, and YDP: TYR1166 to TP1:CYS1.

affinity of −8.5 kcal/mol, six hydrogen bonds, and two cation-pi interactions. Based on the total interaction energy of all the 26 of the TP1-*B. pseudomallei* docked complex structures, the key amino acid residues of TP1 that displayed strong interaction energy with *B. pseudomallei* structures were from the terminal amino acids of peptides; LYS1, ARG5, CYS7, TYR8, ARG9, ILE11, ARG14, CYS16, and ARG17 (Supplemental Information 7).

## DISCUSSION

Treatment of melioidosis involves long courses of antibiotics which often lead to resistance (*Dance, 2014*). Therefore, researchers are aiming for alternative antimicrobial agents such as AMPs in order to contain the situation. AMPs have made a big impact in antimicrobial research, to diminish the inefficacy of antimicrobial therapy in immunocompromised hosts and also ongoing emergence of resistance to conventional antibiotics worldwide (*Giuliani, Pirri & Nicoletto, 2007*). Among the AMPs which

were reported to demonstrate potential to inhibit *B. pseudomallei* include LL-37 (*Kanthawong et al., 2012*), PG1 (*Sim et al., 2011*), bovine lactoferrin (*Puknun et al., 2013*), phospholipase A2 (*Samy et al., 2015*), and SMAP-29 (*Blower, Barksdale & van Hoek, 2015*). However, there are more potential AMPs that are yet to be tested against *B. pseudomallei*. In that instance, TP 1, a 17-residue AMP arranged in two anti-parallel β-sheets connected by two disulfide bonds, isolated from the haemocyte membrane of Japanese horse shoe crab, *Tachypleus tridentatus* (*Nakamura et al., 1988*) was shown to exert broad-spectrum antimicrobial activity against a wide range of Gram-negative (i.e. *E. coli*, and *S. typhimurium*) and Gram-positive bacteria (i.e. *Staphylococcus aureus*) (*Nakamura et al., 1988*; *Ohta et al., 1992*). To date, the inhibition activity of TP1 on *B. pseudomallei* is yet to be reported. As such, our studies focus to understand the mode of action of AMPs, specifically the TP1 and to identify potential interaction targets on *B. pseudomallei*.

In the preliminary screening, we observed that the activity of the AMPs was not affected by the antimicrobial susceptibility of the *B. pseudomallei* isolates. A similar observation was reported by *Mataraci & Dosler (2012)* where their tested strain, methicillin-resistant *S. aureus* (MRSA) ATCC 43300 were susceptible to AMPs indolicidin and cecropin (1–7)–melittin A (2–9) amide (CAMA).

Advancing from the preliminary study, the MIC and MBC values of TP1 were determined to be higher than LL-37 and PG1. These high inhibition values of TP1 towards *B. pseudomallei* may be due to its non-specific binding where more TP1 molecules were required to bind to the *B. pseudomallei* isolates in order to exhibit the inhibition activity. TP1 may be prone to aggregation at inhibitory concentrations where some experimental evidence may assist in reducing the inhibition values of the AMP. The higher *B. pseudomallei* inhibition values may result from the bacteria lowering the membrane surface net charge or altering the hydrophobicity (*Peschel, 2002*; *Poole, 2002*). In addition, the inhibition values of LL-37 and PG1 in our study were higher that the findings of *Kanthawong et al. (2012)* and *Sim et al. (2011)*. This observation may be due to the strain dependant variation among studies whereby *Kanthawong et al. (2012)* used *B. pseudomallei* isolate 1026b while *Sim et al. (2011)* used *B. pseudomallei* isolates from Singapore. Although *B. pseudomallei* K96243 was exposed to LL-37 and PG1, there is always a possibility that the strains used for this study has adapted to the laboratory environment after many passages, thus displaying properties not observed with the same strain used in other laboratories. However, the antimicrobial activity trend in our study was similar and the values obtained from each replicate were the same. In our study, *E. coli* ATCC 25922 was used as a control microorganism since it was reported susceptible to LL-37 (*Barańska-Rybak et al., 2006*), PG1 (*Aumelas et al., 1996*), and TP1 (*Hong et al., 2015*).

In the time-kill study, TP1 inhibited *B. pseudomallei* K96243 within two hours of exposure compared to that of conventional antibiotics, CAZ. Most importantly, TP1 did not enhance *B. pseudomallei* activity. Similar observation was also reported with LL-37 by *Kanthawong et al. (2012)*, where LL-37 inhibited *B. pseudomallei* 1026b, within two hours of post exposure. These observations also support the fact that AMPs exhibit

bactericidal effect at a shorter duration compared to conventional antibiotics, which gives the microorganism lesser chances to develop resistance (*Hancock, 2001*; *Rodríguez-Rojas, Makarova & Rolff, 2014*).

During the biofilm state of the growth, the maximum inhibition of *B. pseudomallei* cells in biofilm state (442 μM) was higher that the MBC of the planktonic cells (221 μM). Similarly, *Anutrakunchai et al. (2015)* have also reported that the CAZ susceptibilities of *B. pseudomallei* biofilm were much higher than those of planktonic cells. Moreover, the protective extracellular polymeric substance encasing *B. pseudomallei* in biofilm state prevents uptake by phagocytic cells (*Taweechaisupapong et al., 2005*) and also causes the bacteria to be refractory to antimicrobial action (*Di Luca, Maccari & Nifosì, 2014*). Due to the non-specific binding of AMPs to the components of the EPS, there was insufficient AMPs molecules to exhibit inhibition effect on the bacterial cells. In a study by *Chan, Burrows & Deber (2004)*, *Pseudomonas aeruginosa* in biofilm form secrete alginate which induces α-helical conformation on modified magainin 2 and cecropin P1, thus diminishing its antimicrobial activity. Based on our data, there is always a possibility of *B. pseudomallei* AMP resistance in biofilm state but TP1 has shown reproducible results across replicates, comparable to that of PG1 and possibly even more consistent than LL-37. Moreover, both TP1 and PG1 were observed to inhibit *B. pseudomallei* in planktonic and biofilm state. However, LL-37 only exerted its antimicrobial activity on planktonic *B. pseudomallei* cells. Our observation contradicted with the study conducted by *Kanthawong et al. (2012)*, where they reported anti-*B. pseudomallei* biofilm activities with 100 μM of LL-37 and biofilm inhibition activities with 936 μM of CAZ. This could be due to the strains variation, where strains resistant to CAZ but susceptible to LL-37 were used, as compared to the *B. pseudomallei* K96243 used in our study (susceptible to CAZ and tolerated LL-37) in biofilm state. In addition, CAZ and MRP appeared most active against *B. pseudomallei* K96243 in biofilm state as the isolate was already susceptible to both the antibiotics from the antibiotic susceptibility profile (Supplemental Information 2). Where serial dilution and plating on NA was involved, the CFU/ml or CFU/biofilm can only be detected if there were any bacteria colonies growing on the agar surface. The graph in Fig. 3 plateaus at approximately 4 log CFU/biofilm was an example where the limits of detection may apply. Therefore, in these studies, although countable colonies are present but below the countable range (standard counting range are between 25–250 colonies), they were still counted and reported.

Overall, in the MIC and MBC method used in our study slightly differs from the CLSI methods for measuring antibiotic susceptibility (*Cockerill et al., 2012*) . RPMI 1640 was used as it is able to sustain bacterial growth although not as nutrient-rich compared to LB or Muller Hinton broth (MHB). *Schwab et al. (1999)* used RPMI 1640 (with 20 mM HEPES and without sodium bicarbonate) to study the inhibition activities of AMPs D2A21 and D4E1 on *S. aureus* ATCC 29213 and *P. aeruginosa* ATCC 27853. The highest AMPs activity in this media was observed, while it supported the growth of the bacteria, as compared to trypticase soy broth and nutrient broth. We speculate that the extracts in LB (yeast) and MHB (beef) do not have a consistent composition may contain

components which may lead to the non-specific bindings of the AMPs and thus diminishing the AMPs activity.

Our SEM analysis revealed that TP1 acts on the bacterial membrane, suggesting that due to the loss of membrane integrity, the bacteria fail to contain their cytoplasmic components and thus, causing them to lyse. The observations with *E. coli* were similar to the findings of *Hartmann et al. (2010)* with *β*-hairpin AMPs. To the best of our knowledge, this is the first report on the effect of TP1 on *B. pseudomallei*. Furthermore, also using *E. coli* ATCC 25922, *Hong et al. (2015)* also observed that TP1 damaged the entire cell, including the structure of the cell wall and membrane. Pore formation and partial disruption of the cell wall and cytoplasmic membrane was also observed, resulting in the outflow of cell contents and the complete collapse of some cells.

The AMPs in this study were synthesized as acetate salts which do not pose any known effects as the presence of trifluoroacetic acid (commonly used in peptide synthesis) is cytotoxic and undesirable in preclinical and clinical studies (*LifeTein, 2015*). At the moment, TP1 is not suitable for therapy due to its in vitro cytotoxicity unless certain modification was done (i.e. to reduce non-specific binding). As limited data is available to modify the AMP; we believe that our findings will contribute to the existing literature so that this AMP can be modified to specifically target either bacteria cells or cancer cells. Although LL-37 was reported to possess anti-*B. pseudomallei* properties (*Kanthawong et al., 2012*), it has been reported to be cytotoxic to both A549 (*Aarbiou et al., 2006*), and AGS (*Wu et al., 2010*) and was not included in our study. On the other hand, PG1, was reported cytotoxic against Hep G2 (*Niu et al., 2015*) but the effect was yet to be reported on A549 and AGS. One of the disadvantages of AMPs is they are cytotoxic will require certain modification to nullify the effect.

In the in silico study, we have successfully docked TP1 on the LPS of *E. coli* with similar parameters as stated in *Kushibiki et al. (2014)* using ADV instead of AutoDock 4.2 where ADV was programmed with significant improvement the average accuracy of the binding mode predictions compared to AutoDock 4.2 (*Trott & Olson, 2010*). Our findings were similar to *Kushibiki et al. (2014)* where we observe that ARG17 residues of TP1 were involved in the binding to LPS while the PO4 groups of the LPS were involved in the binding to TP1 (hence the hydrogen bonds between both molecules). Moreover, we also observed that the lack of non-covalent interactions (pi-interactions), maybe due to the lack of benzene and other aromatic residues in the LPS molecule to contribute to the interaction. The slight differences in the binding residues of TP1 to LPS maybe due to the differences in the algorithm of both docking software (*Trott & Olson, 2010*). However, most of binding residues in the LPS of *E. coli* in the docking study by *Kushibiki et al. (2014)* were reproducible. On the side note, we are aware of the variation of the O-antigen (connected to the LPS in the outer membrane) among Gram-negative bacteria. However, the basic structure of the LPS comprised of repeated oligosaccharide units, where each unit was made up of common sugars such as hexoses and hexosamines (*Reyes et al., 2012*). Therefore, interaction of TP1 with *E. coli* may be used as a basis for the prediction of the TP1 interaction with *B. pseudomallei*. At the moment, the exact role of the YDP of *B. pseudomallei* has yet to be reported. However, from

the verification of the homology modelled *B. pseudomallei* protein based on PspA, and the membrane blebbing observed in the SEM analysis, we hypothesize that the YDP may contribute to a surface protein on *B. pseudomallei* where it interacted with TP1 molecules.

Taken together, our docking results show negative binding energies indicated favourable bindings of TP1 with all three receptors, LPS of *E. coli* and the homologically modelled YDP. Although the binding affinity of TP1 to the LPS was not as strong as that of autolysin, pneumolysin, and PspA, the IE of TP1-LPS complex was the most negative, indicating a stronger interaction as compared to PspA autolysin, and pneumolysin. Moreover, the binding affinity and IE of TP1 to the autolysin, pneumolysin, and PspA may indicate that the binding sites for the respective molecules were the common binding sites for *S. pneumoniae* inhibitors.

The binding affinity and the IE of YDP model was higher than that of the template molecule PspA indicating that TP1 has a stronger interaction with the YDP model. Moreover, the IE of YDP model was observed to be similar than that of LPS of *E. coli*, which hinted that TP1 has similar interaction strength for both of the molecules. However, there were no pi-interactions in the TP1-YDP complex as compared to the TP1-PspA complex. This may indicate that TP1-PspA complex was more stable compared to TP1-YDP complex but TP1 has a stronger interaction to YDP model compared to PspA complex.

Based on our docking study with *B. pseudomallei* PDB structures, TP1 has the highest probability of binding to *B. pseudomallei* cycle inhibiting factor (PDB ID: 3GQM) when both molecules are in close proximity since the total interaction energy was the most negative of all the docked proteins. The binding of TP1 to the cycle inhibiting factor of *B. pseudomallei* is expected cause the down regulation of this protein during the interaction. Similar down regulation of the universal stress protein was reported by *Aanandhi et al. (2014)* on a study of the interaction natural polyphenols with *Mycobacterium tuberculosis*. Moreover, the binding of AMPs to proteins in general are based on a few key residues on the AMPs themselves. These are often termed as "hot spot" residues (*Clackson & Wells, 1995*; *London, Movshovitz-Attias & Schueler-Furman, 2010*). In our study, we have identified key residues in TP1 which react with all the *B. pseudomallei* proteins, summarized and calculated based on their critical contribution to the total binding energy. As such, we propose that any modification done on TP1 specifically for *B. pseudomallei* should avoid tempering with those key residues.

## CONCLUSIONS

Herein, we have selected commercially available peptide, TP1 as the potential AMP against *B. pseudomallei*. Based on the above findings, TP1 has shown great prospect as an anti-*B. pseudomallei* agent where its efficacy is comparable to that of LL-37 and PG1. The in silico data suggest that TP1 has a strong interaction not only to *B. pseudomallei* proteins but also other Gram-negative and Gram-positive bacteria as well. Besides that, we have demonstrated TP1's efficacy on *B. pseudomallei* isolated in both planktonic and biofilm form with the additional supporting data from the in silico study. Possible TP1 interactions with the common peptide or inhibitor binding targets for LPS of *E. coli*, as

well as autolysin, pneumolysin, and PspA of *S. pneumoniae* from in silico molecular docking study were also identified. Further modifications of TP1 can be done to enhance its specificity to *B. pseudomallei* and to reduce its cytotoxicity. With the current technology, we also propose further experiments such as coarse-grain molecular dynamics to simulate the interaction between TP1 and *B. pseudomallei* outer membrane (*Hall, Chetwynd & Sansom, 2011*). Besides that, TP1 can also be tagged with a fluorescent compound to visualize TP1 movement in live *B. pseudomallei* cells (*Gee et al., 2013*). Proteomic analysis and embedding of TP1 on resins to carry out affinity chromatography on *B. pseudomallei* whole bacteria, coupled with liquid chromatography-mass spectrometry will enable in-depth understanding on the effect of TP1 on *B. pseudomallei* protein structures (*Casey, Coley & Foley, 2008*; *Ortiz-Martin et al., 2015*).

## ACKNOWLEDGEMENTS

We would also like to thank Dr. Ding Jeak Ling and Dr. Roger W. Beuerman (National University of Singapore), and Dr. Bob Hancock (University of British Columbia) for providing the AMPs for this study.

### Funding

This work was supported by the Ministry of Higher Education (MOHE) Malaysia under the High Impact Research (HIR)–MOHE project UM.C/625/1/HIR/MoE/CHAN/02 (H-50001-A000013), the Ministry of Science, Innovation and Technology (MOSTI), the Malaysia under the Science Fund (55-02-03-1002), and the University of Malaya Research Grant (UMRG RP027B-15AFR). The funders had no role in study design, data collection and analysis, decision to publish, or preparation of the manuscript.

### Grant Disclosures

The following grant information was disclosed by the authors:
Ministry of Higher Education (MOHE) Malaysia under the High Impact Research (HIR)–MOHE project UM.C/625/1/HIR/MoE/CHAN/02: H-50001-A000013.
Ministry of Science, Innovation and Technology (MOSTI), Malaysia under the Science Fund: 55-02-03-1002.
University of Malaya Research Grant: UMRG RP027B-15AFR.

### Competing Interests

The authors declare there are no competing interests.

### Author Contributions

- Lyn-Fay Lee conceived and designed the experiments, performed the experiments, analyzed the data, wrote the paper, prepared figures and/or tables.
- Vanitha Mariappan reviewed drafts of the paper.
- Kumutha Malar Vellasamy reviewed drafts of the paper.

- Vannajan Sanghiran Lee contributed reagents/materials/analysis tools, reviewed drafts of the paper, funding for in silico work.
- Jamuna Vadivelu conceived and designed the experiments, contributed reagents/materials/analysis tools, reviewed drafts of the paper, funding for in vitro work.

## Data Deposition

The raw data has been supplied as Supplemental Dataset Files.

## Supplemental Information

Supplemental information for this article can be found online at http://dx.doi.org/10.7717/peerj.2468#supplemental-information.

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
