# Peer review of "Antimicrobial activity of Tachyplesin 1 against Burkholderia pseudomallei: an in vitro and in silico approach"

_PeerJ, doi:10.7717/peerj.2468_

## Round 0.1 · original submission · Major Revisions

The reviewers raise a number of important points which must be addressed in a revised manuscript. These include potential problems with your calculations and some experimental protocols, and a lack of justification for following TP1 as lead, especially given its toxicity to mammalian cells. Another major criticism of one of the reviewers was the limited in silico analysis performed, and the lack of any experimental confirmation of the results. This gives me the impression that this part of the work is very preliminary. This, and the toxicity of TP1, also leave me feeling somewhat mislead by the title of your manuscript, which must be addressed if you choose to revise your manuscript.

·

Basic reporting

English is in need of tightening in some places. Some sections could be more succinct. The abstract is a good example.
The in silico work is not adequately introduced (for me, a non-expert in that field).
Figure 1 is not needed, data is in the text. Legends need more information so the individual figures are stand alone.
I think there are two studies, first the lab experiments to justify and underpin in silico studies (which I am not convinced that they do), and second the in silico experiments following a clearly justified strategy (which is not apparent in this draft).

Experimental design

The strategy for screening against clinical isolates, testing anti-biofilm effects and cytotoxicity is OK. The choice of TP1 for further study is not well justified. The in silico section does not have a strategy justifying the validity of the experimental design, and the in silico studies research question is not clearly stated and definitely not aligned to the stated aim of the manuscript in general: AMPs as alternative therapies in the background of resistance to conventional antibiotics. Plus other comments in the comments to authors.

Validity of the findings

I think there are some basic errors in the research (possibly at calculation or interpretation stage) that are concerning; e.g. 10e12+ CFU per ml in a microplate well culture, stating a reduction of 7.8 log CFU/ml to 4.2 log CFU/ml is a 2 fold reduction. Plus other comments in the comments to authors.

Additional comments

The development of antimicrobial resistance is a serious public health concern. Some bacteria, e.g. Burkholderia pseudomallei also have high levels of intrinsic resistance to antibiotics. The development of further resistance in B. pseudomallei means there are some strains now resistant to many previously usable antibiotics, and increasingly for some people meliodosis does not respond to antibiotic treatment. Antimicrobial peptides (AMPs) are a good alternative candidate to conventional “antibiotics”. The manuscript under review examines AMPs as a potential treatment for meliodosis by first screening the activity of a range of AMPs against a range of B. pseudomallei isolates. LL-37 and PG1 show best range and lowest MICs, but TP1 is chosen for further study without convincing justification. No comparison of AMP activity profiles is made to any data that might be available for the isolates tested (e.g. API20 NE profile, clinical presentations, antibiotic sensitivity profile). Anti-biofilm activity is tested as an expansion of the data, this is a good strategy but there are serious problems with the presentation and interpretation of this data (see specific comments). SEM is used to begin to give some gross mechanism data, but the evidence presented does not really support the conclusions made. In vitro toxicity testing with a range of cell lines suggests that TP1 is more toxic to mammalian cells than B. pseudomallei, but no consideration is given to how this finding may suggest that TP1 is unsuitable for therapeutic use for meliodosis.
A substantial section is devoted to in silico analysis of TP1 and identification of proteins that TP1 may interact with. I do not consider this my area of expertise, but make some high-level comments in the specific details. I do feel the manuscript is long enough without this information and this section would be best left out and used as the basis of a separate manuscript.
Overall, the figures presented need legends that supply information to make the figure stand alone. For example the concentrations of AMPs used must be stated clearly in the legend. I do not believe that supplementary information is appropriate for an online journal – it is either needed for the paper (include it) or not (leave it out).

Specific comments.
Abstract.
Too long and rambling. It needs to be re-written to be more succinct and highlight findings.
Introduction.
Line 64. What do the authors mean by “phenotypically mutate”. Is biofilm formation more correctly described as phenotypic adaptation, without necessarily requiring any mutation, which I would class as a genotypic change.
Line 71. What is meant by “inoculation” – can this mode of dissemination be described.
Line 86. I think “synthetic” rather than “synthesized” better describes the authors’ intent here.
Line 88. The sentence starting “They are also highly potential …” doesn’t make sense to me. Perhaps highly potent?
Line 90. Begin the sentence with Groups of …
The overall strategy of the in silico approach to drug design here needs to be clearer; i.e. identify something that works and use an in silico approach to generate hypothesis for mechanism of action and new designs for better acting drugs. Specific examples where this approach has been successful should be briefly described.
Methods.
Peptide storage and Preliminary screening. Please indicate the concentrations of stocks and tests for the preliminary screening. It would also be helpful to include molar concentrations and g/l concentrations for each.
MIC and MBC of planktonic cells. Why was RPMI used as the medium? The methods do not look like internationally accepted methods for measuring MIC or MBC that would be used for “antibiotics” so should be given a brief justification. Dilution of the cells in RPMI will give a final concentration of 0.8x RPMI, which should be commented on. The same comments apply to the testing of anti-biofilm activity.
Please indicate more than the range of concentrations, i.e. doubling dilutions for 200microM? Or state the actual concentrations tested. . The same comments apply to the testing of anti-biofilm activity.
Assays were performed in a U-shaped microplate, followed by measurement of absorbance at 570nm. I think flat bottomed microplates would be needed for Absorbance readings. How was the MIC decided upon, i.e. how much inhibition was needed?
Please explain exactly how the time to kill curve was performed, first I don’t see how A570 (as stated in the manuscript) can be used for this, unless there is a high inoculum and reduction in absorbance due to cell lysis is measured. Plating should be used, and the Sieuwerts method stated used plating- but how much medium was used?, I think the experiment states taking 24 x hourly readings from 0.1ml total volume.
Results.
Preliminary screening.
Figure 1 is not needed, the information is in the text. For TP-1 83/100 strains were susceptible, but this was calculated at 83.33% (it is 83%). The use of 100.00% also is 2 too many decimal places. Please also indicate here or in the methods the criteria for classification as sensitive.
No comparison of AMP activity profiles is made to any data that might be available for the isolates tested (e.g. API20 NE profile, clinical presentations, antibiotic sensitivity profile). This should be done, even if there is no correlation observed, and this may only be possible for TP1 where there are sensitive and insensitive isolates.
Please justify selection of TP1 as the AMP chosen for further study, LL37 and PG1 hit all the isolates and are more potent (lower MIC/MBC).
Time to kill.
The section repeats the results description, please rewrite to be more concise. Figure 2 is of concern, the untreated reaches a concentration of 10e12 or 10e13 cfu/ml, and what the methods suggest is in a microplate well. My experience is that this number of cfu may be obtained from a litre of bacteria grown overnight in a rich medium, not per ml in 0.8x RPMI in a microplate. I suggest checking calculations and if they are correct then the method used has some deficiency or artefact. I am happy with the data being used for time to kill (with limits of detection) based on those platings that do not grow bacteria, but not with the enumeration of cfu, which is clearly incorrect.
The comparison with CAZ, a drug in clinical use against B. pseudomallei is a good control. Are the amounts CAZ used clinically relevant?
Anti-Biofilm activity.
The presentation and interpretation of results needs to be thought about and redone. The first line states, The number of … biofilm forming cells (7.84 log CFU/ml) was reduced by two fold (4.2 log CFU/ml)…. This is almost a 3 log reduction in CFU! (1000 fold reduction). I would also use CFU per biofilm in the assay well, not the concentration in the suspension released from the biofilm.
In figure 3 DJK5 is *’d with a significant effect, but the graph does not support this. If there is a reduction it is very small. An important set of findings here concern LL37 and PG1, and CAZ and MRP. LL37 is not active against the biofilm, while PG1 appears more active than TP1, and should be discussed regarding the choice of AMP to study further. CAZ and MRP, the classical antibiotics appear most active, this should be discussed and the experiment should be repeated with isolates resistant to these antibiotics to demonstrate benefit of the AMPs. Fig 4 should be discussed with correlation to planktonic MIC/MBCs, MIC planktonic is 221 microM, maximum anti-biofilm effect is seen at 442 microM. Also looking at the anti-biofilm effect here the limit of detection (LoD) of the test should be indicated, I wonder why the graph plateaus 1x10e4 CFU per ml? How many colonies on the plate does this correspond too? And are we at the LoD? Fig 5 looks at the key treatments, but values differ from Fig 3. The legends really need to state the concentrations of each antibiotic and AMP used. In Fig 5 LL37 is *’d, but this is an increase in CFUs and should be highlighted as such. I would recommend highlighting (and testing for) the treatments that reduce bacterial numbers.
SEM
A small number of cells in one image per treatment is given. Only 2 cells are in 6A. If any conclusion is to be made on cell dimensions this needs to be on a substantial number of cells, e.g. 6 fields of view containing >50 cells and from three independent experiments. The data should then be graphed, and analysed with appropriate statistics.
Line 290. I’m not certain on the statement that cells are 5 micro m in length. In Fig 6B there look like pairs of cells that have divided, but are still attached, and here the pair is not 5 micro m in length. It may be that complimentary techniques, e.g. using a membrane stain and fluorescence microscopy, is needed to confirm this.
Line 290. Where there are references made to specific features in an SEM image these should be indicated with e.g. an arrow.
Figs 6B and D. Is the “debris”, debris of killed cells and are they blisters and bubbles on the cells? Or maybe it is aggregating protein? Perhaps an image set of an AMP and a resistant isolate would help as a comparison?
Overall, the interpretations from the SEM are not supported by the data presented.
Cytotoxicity.
In vitro toxicity testing with a range of cell lines suggests that TP1 is more toxic to mammalian cells than B. pseudomallei. What does this mean for the therapeutic use of TP1, as alternative therapies are the proposed aim of the study? Perhaps this should also be compared to LL37 and PG1, where there is data in the literature already published showing at least for LL37, and it is more toxic to bacterial cells than mammalian cells. The discussion (e.g. section beginning Line 465) considers action as an anti-cancer agent, which is an aside that does not address the key findings that suggest TP1 will be toxic to human cells at doses that are anti-bacterial and anti-biofilm. Armed with these toxicity results, would an animal or human ethics committee approve animal experiments or human trials? I think the answer would be no.
In silico molecular docking.
This is not my area of expertise, and I am not qualified to comment on the specific methodology. The actual proteins interacting with TP1 are not clearly identified. I would be more overt in reference to Figure 3. I feel this table would also benefit from predicted or known cellular location of the target protein (i.e. are they surface exposed). I did struggle to see the strategy justifying the in silico experiments, especially as this could easily be tailored to the design of an anti-meliodosis agent that is potent against B. pseudomallei while not affecting human cells at therapeutic concentrations. In addition the relevance of AMP binding to proteins is only addressed at the end of the discussion, as possibly a route to aiding the clearance of bacterial lysis products. So, again I wonder if the strategy is viable and would like to see this clearly explained.

Reviewer 2 ·

Basic reporting

English usage and appropriate use of spaces throughout needs editing and polishing. Too many examples to list them all, but see for example, line 88 ("also highly potential"), line 90 ("There" should be three), line 92 ("several" should be few), line 103 ("give" should be "gives"), line 113 ("MDR" is not a thing to combat. Combat bacteria or drug resistant bacteria, not MDR).
FIrst paragraph of Introduction is not necessary - delete. No issues of antibiotic resistance are dealt with in this paper.
Second paragraph of Introduction should begin with "Burkholderia pseudomallei" as the first paragraph of the introduction.

The major issue with this work is that it does not meet the criteria of representing "an appropriate ‘unit of publication’, and should include all results relevant to the hypothesis." The work begins with some in vitro antimicrobial assays, and then a computational prediction of binding of the peptide to 26 Burkholderia proteins is presented, but no experimental confirmation or validation of the predicted interactions is presented. Thus, this part of the paper is fundamentally incomplete and does not prove the hypothesis.

Screening results at 1 mg/ml peptide against the bacteria (line 392) suggests that the peptide is not very active indeed. Also this concentration is not mentioned in the methods (line 150) and should be there and indicated in Figure 1 legend. What is the molarity of this concentration? These experiments are very far from the reported MIC levels (62 uM). It would also be of interest to list in the supplemental the 20 strains that were resistant to TP1 and some analysis of why that might be.

Experimental design

This work fails to meet the standard in two places.
1) Anti-biofilm assays are presented. However, this experimental approach and the authors data shows that the peptides aggressively kill the bacteria within the biofilm. Thus there is no direct antibiofilm effect, only a bactericidal effect as was previously shown in the paper, and as a result of few bacteria remaining, there is less biofilm produced. This is not an appropriate understanding of the concept of an antibiofilm peptide or an antibiofilm approach. It would be acceptable to state that the peptide can kill bacteria that are in a biofilm, but that does not mean that TP1 is an "antibiofilm" peptide as currently is written. "Antibiofilm" is a direct regulation of biofilm production in live bacteria, not the killing of bacteria. Ceftazdamine is not antibiofilm, it is 100% bactericidal. These experiments are more appropriately performed at sub-MIC concentrations to account for this issue.
2) There is no validation of the purely computational prediction of binding of the peptide to 26 Burkholderia proteins is presented. No experimental confirmation or validation of the predicted interactions with Bp cycle inhibiting factor is presented. Thus, this part of the paper is fundamentally incomplete and does not prove the hypothesis. Also, this analysis severely under-represents the totality of proteins in Burkholderia, only testing 26. The odds of one of those 26 proteins being a true target of the proposed peptide is very unlikely. No positive control (a protein known to interact with an antimicrobial peptide) is presented for comparison to know if the cutoffs proposed by the authors have any biological relevance.

Validity of the findings

There is no validation of the purely computational prediction of binding of the peptide to 26 Burkholderia proteins is presented. No experimental confirmation or validation of the predicted interactions is presented. Thus, this part of the paper is fundamentally incomplete and does not prove the hypothesis. Experimental association of the proposed protein with the peptide needs to be demonstrated to prove the computational prediction.

Also, this analysis severely under-represents the totality of proteins in Burkholderia, only testing 26 out of the 6000-7000 CDS in B. pseudomallei. The odds of one of those 26 being a true target of the proposed peptide is very unlikely. No positive control (a protein known to interact with an antimicrobial peptide) is presented for comparison to know if the computed interaction value cutoffs proposed by the authors have any biological relevance. A computational demonstration of another known protein-peptide interaction should be run as a comparison. For example, LL-37 was recently shown to interact with a bacterial protein AcpP and there is a known interaction of LL-37 with the P2X7 receptor. Other cathelicidins can interact with Lpp on Pseudomonas which could also be modelled.

---

## Round 0.2 · Major Revisions

As you will see from the reviewer's comments, they are happy with the progress you have made but in their opinion there are still major revisions required before the paper can be accepted.

·

Basic reporting

The tracked changes document is unhelpful- there are no line numbers and the changes are not tracked, but the area where there is a change is highlighted. This makes re-review difficult. Figure legends are not in this document.

Experimental design

see annotations (green type) in responses to author's rebuttal letter.

Validity of the findings

see annotations (green type) in responses to author's rebuttal letter.

Additional comments

see annotations (green type) in responses to author's rebuttal letter.
Overall the authors have made a creditable effort to address a number of criticisms, for some there is still work to do.

---

## Round 0.3 · Major Revisions

I'm sorry, but the reviewer still does not feel your manuscript is suitable for publication. You will need to address their concerns before your manuscript can be accepted.

·

Basic reporting

The authors have made significant effort to describe a multidisciplinary project in English and make it accessible to all readers. Areas could be improved, but it is at an acceptable level now.

Experimental design

No comments beyond my uploaded document, which makes a couple of minor points.

Validity of the findings

The uploaded document expands my comments here. I don't believe the cfu/ml values quoted and these need to be demonstrably fixed -i.e. numbers on graphs above 10e10 need additional data to support, as do any changes to normally obtained cfus/ml, before the manuscript is publishable.

Additional comments

see uploaded response to rebuttal.

---

## Round 0.4 · Major Revisions

Thank you for taking the time to repeat the killing assay and uploading the data. I have had a look at this latest data, and agree with the reviewer - it is physically impossible for the number of bacteria you have calculated to fit into the volume of the wells. A culture of E. coli that is 10^15 CFU per ml would be 10^14 CFU in 100ul. Taking an E. coli cell as having a volume of 1.3 cubic um, this means that the amount of bacteria you describe would take up the space of 0.1L, 1000x more than the space available. Using such a calculation, the maximum number of cells that would fit in 100ul is 10^11 CFU. This is inline with my and the reviewers experience with E. coli. I have also consulted another researcher working with Burkholderia pseudomallei and they agree that the bacteria will not grow to numbers higher than 10^11 CFU in vitro.

This suggests to me that there is a problem with your dilution series, in that the 10-fold dilutions you report are less than 10-fold dilutions. Such errors tend to be amplified when doing dilutions with small volumes, and then plating out small volumes. Are you using a multi-channel pipette and changing the tips between each dilution?

I cannot accept a manuscript that includes numbers that are physically impossible to reach. In order to accept this manuscript for publication, you will need to investigate where your errors are coming from and then repeat the experiments to give their true values.

---

## Round 0.5 · accepted · Accept

Thank you for addressing the concerns the reviewers and I had over the methodology.

·

Basic reporting

The authors have made significant effort to describe a multidisciplinary project in English and make it accessible to all readers. Areas could be improved, but it is at an acceptable level now.

Experimental design

All previous comments have been addressed.

Validity of the findings

The cfu/ml values quoted are now in the believable range and I'm satisfied that the technical issues have been addressed in the new data now in the manuscript.